# New Insights in the Setting of Transplant Oncology

**DOI:** 10.3390/medicina59030568

**Published:** 2023-03-14

**Authors:** Silvia Quaresima, Fabio Melandro, Francesco Giovanardi, Kejal Shah, Valerio De Peppo, Gianluca Mennini, Davide Ghinolfi, Ashley Limkemann, Timothy M. Pawlik, Quirino Lai

**Affiliations:** 1General Surgery and Organ Transplantation Unit, AOU Policlinico Umberto I, Sapienza University of Rome, 00161 Rome, Italy; 2Department of Surgery, Division of Surgical Oncology, James Cancer Center, The Ohio State University, Columbus, OH 43210, USA; 3Hepatobiliary Surgery and Liver Transplantation, University of Pisa Medical School Hospital, 56124 Pisa, Italy

**Keywords:** peri-hilar cholangiocarcinoma, intrahepatic cholangiocarcinoma, colorectal liver metastases, neoadjuvant chemotherapy, liver transplantation

## Abstract

*Background and Objectives*: Liver transplantation (LT) is the best strategy for curing several primary and secondary hepatic malignancies. In recent years, growing interest has been observed in the enlargement of the transplant oncology indications. This paper aims to review the most recent developments in the setting of LT oncology, with particular attention to LT for unresectable colorectal liver metastases (CRLM) and cholangiocellular carcinoma (CCA). *Materials and Methods*: A review of the recently published literature was conducted. *Results*: Growing evidence exists on the efficacy of LT in curing CRLM and peri-hilar and intrahepatic CCA in well-selected patients when integrating this strategy with (neo)-adjuvant chemotherapy, radiotherapy, or locoregional treatments. *Conclusion*: For unresectable CCA and CRLM management, several prospective protocols are forthcoming to elucidate LT’s impact relative to alternative therapies. Advances in diagnosis, treatment protocols, and donor-to-recipient matching are needed to better define the oncological indications for transplantation. Prospective, multicenter trials studying these advances and their impact on outcomes are still required.

## 1. Introduction

Liver transplantation (LT) is the best strategy for curing several acute and chronic liver diseases, including certain primary and secondary hepatic malignancies [1]. Hepatocellular carcinoma (HCC) in both cirrhotic and non-cirrhotic livers is a well-established indication and remains the most common oncologic indication for LT [2,3]. According to the most recent published studies, HCC patients present excellent post-transplant survival rates, with 5-year overall survival (OS) rates >80–85% and recurrence rates <10–15% [2].

Less common primary cancers in selected patients considered eligible for LT include peri-hilar cholangiocarcinoma (pCCA) and hepatoblastoma in pediatric patients [4]. Well-selected patients with unresectable liver neuroendocrine tumor metastases are also considered potential candidates for LT [5].

With significant advances in chemotherapy, surgical techniques, and immunological therapies, there has been an extension of oncologic indications for liver transplantation in recent years. In 2015, Dr. Lerut first introduced the concept of “transplant oncology” to describe the intersection of oncologic management and liver transplantation [6]. Several centers have been examining the role of LT for unresectable intrahepatic cholangiocarcinoma (iCCA) and unresectable colorectal liver metastases (CRLM), as well as extending the indication for HCC. This paper aims to review the most recent developments in liver transplant oncology, with particular attention to liver transplant for colorectal metastases and cholangiocarcinoma.

## 2. Liver Transplantation for Cholangiocarcinoma

### 2.1. Perihilar Cholangiocarcinoma

Perihilar cholangiocarcinoma (pCCA) is the most frequent type (50–70%) of extrahepatic cholangiocarcinoma [7]. Due to an often initially asymptomatic clinical course and a highly aggressive nature, patients with pCCA often present with advanced disease with limited curative treatment options. Complete (R0) surgical resection is the primary treatment option to provide long-term survival for resectable pCCA [8]. However, even with R0 resection, reported long-term OS remains low (5-year: 15–35%) [9,10].

LT represents the only potentially curative treatment option for patients with unresectable pCCA. Primary sclerosing cholangitis (PSC) is a well-known risk factor, with nearly 30% of PSC patients developing cholangiocarcinoma. Among patients who develop pCCA with concomitant chronic liver disease such as PSC, LT provides complete tumor removal and a cure for the underlying chronic liver disease [11].

Historical attempts to treat pCCA with LT led to unacceptable rates of disease recurrence [12,13]. Meyer et al. and Seehofer et al. reported 3- and 5- year OS reaching 40% and 30%, respectively [14,15]. Of note, survival was not improved when LT was combined with pancreaticoduodenectomy, although higher rates of R0 resection were obtained [16,17]. Recurrence rates reported in studies published before 2000 were disappointing. Such poor results were likely due partly to limitations in the design of the studies. For example, in several reports, survival among patients following LT who had pCCA were combined with the results of patients who had iCCA [14,15,18]. Furthermore, there was significant heterogeneity in many case series regarding disease extent, presence of metastatic lymph nodes, primary liver disease (PSC vs. de-novo CCA), and use of (neo)-adjuvant therapy administration protocols. This heterogeneity is particularly relevant considering that several groups identified tumor size and nodal involvement as predictive factors for pCCA recurrence [18,19,20].

In 2002, the University of Nebraska reported improved survival among patients enrolled in a pretransplant chemoradiation protocol followed by LT as a curative option for pCCA [21]. The Mayo Clinic expanded on these findings in 2004 by proposing a pre-transplant protocol that included neoadjuvant administration of external beam radiation therapy combined with high-dose intravenous 5-fluorouracil (5-FU) followed by intraluminal iridium brachytherapy and oral capecitabine (Table 1) [22]. This protocol, known as the “Mayo protocol”, involves a strict selection process. Patients with radiological evidence of nodal disease, metastasis, or tumors >3 cm in diameter are excluded from LT consideration. Following neoadjuvant therapy, operative staging, including hilar node sampling (regardless of appearance), is performed. Staging must be performed close to the time of LT to confirm an appropriate indication for LT.

The early published results from the Mayo Clinic were encouraging, with 5-year OS at 82% versus 21% among patients undergoing LT versus partial hepatectomy in lymph-node-negative localized disease [22]. Using the same protocol, the Mayo Clinic reported on a larger cohort in 2008 and noted a 5-year survival of 71% [32]. Subsequently, the Mayo protocol was validated in North America in a multicenter study that included 12 centers (n = 287), with a reported 5-year disease-free OS of 65% [26]. In Europe, Mantel et al. investigated the results of LT for pCCA in a European Liver Transplant Registry (ELTR) cohort; these investigators demonstrated a 5-year OS of 59% among patients transplanted following the Mayo protocol criteria [27]. Patient selection and neoadjuvant therapy are likely the most critical factors that improved survival when using the Mayo protocol [27,33]. Of note, the Mayo protocol has been applied to treat pCCA in PSC patients and individuals with unresectable de novo pCCA. Patients within the PSC group can sometimes be unsuitable for resection due to multifocal pCCA or underlying liver disease. Thus, LT generally represents the most suitable option for these patients. In addition, post-LT survival outcomes are often more favorable among patients with PSC versus patients with unresectable de novo pCCA [34,35].

Based on the evidence supporting LT, standardized model for end-stage liver disease (MELD) exception scores have been approved by UNOS/OPTN since 2009 for qualifying patients with unresectable pCCA. In this way, patients may receive the same exception score as candidates with HCC [36]. However, despite using MELD exception points, patients with pCCA have a lower chance of obtaining an LT than individuals listed for other indications. Variable pre-LT dropout rates due to either disease progression, intolerance to neoadjuvant therapy, positive staging, or death have been reported. Rea et al. reported a 5-year OS of 82% among patients who underwent LT, but survivals decreased to 58% in the intention-to-treat analysis due to 46% of the enrolled patients not reaching transplant [11]. In a different study, Ethun et al. reported a much lower dropout rate of 25% based on multicenter data, highlighting the variability in reported dropout rates [29].

Prior to initiating the Mayo protocol, pathological confirmation of pCCA was required. However, a pathologic diagnosis in patients with suspected pCCA can sometimes be challenging. Benign strictures, particularly in patients with PSC, often mimic malignancy, and it can be difficult to differentiate these two conditions based only on clinical and imaging examination [37]. For instance, endoscopic biopsy and brushings are positive in only approximately 30% of cases; then, transabdominal biopsy automatically excludes patients from transplant consideration due to concerns for tumor seeding according to the standard MELD exception points for pCCA.

A recent meta-analysis based on 20 studies examining LT for pCCA reported a 5-year OS that exceeded 50% among LT patients who completed neoadjuvant chemoradiation protocol before transplant; in contrast, OS dramatically decreased to 31.6% among patients who directly proceeded transplantation [38]. Neoadjuvant chemoradiation significantly reduces the risk of tumor recurrence versus upfront transplantation (51.7% with neoadjuvant therapy vs. 24.1% without neoadjuvant therapy) [38,39,40]. As recurrence is the primary cause of death following LT among pCCA cases [39], US centers strongly recommend avoiding LT for pCCA without neoadjuvant chemoradiation [29,38]. In contrast, no European transplant centers include neoadjuvant chemoradiation as a prerequisite for graft allocation in LT for pCCA. More extensive data on disease relapse in transplant patients who do not undergo neoadjuvant therapies are required to definitively understand the role in preventing post-LT recurrence. On this point, the ISO score allocation policy, applied in Italy from 2016 onward, allows for the use of 5% of transplantable grafts for extended purposes, which fscilitates the exploration of new transplant oncology indications within internationally recognized protocols [41]. Therefore, under this allocation policy, pCCA can be transplanted in Italy within the Mayo protocol criteria.

Several retrospective studies have evaluated the potential role of LT for initially resectable patients. Based on these studies, as summarized in Table 2, it is unclear whether the survival difference between transplantation and resection for resectable pCCA should be considered significant enough to justify transplantation [42,43]. Croome et al. published a retrospective analysis from the Mayo Clinic comparing de novo pCCA patients who underwent resection versus neoadjuvant therapy plus LT. There was improved 1-, 3-, and 5-year OS among patients who underwent LT (90%, 71%, and 59%, respectively) versus resection (81%, 53%, and 36%, respectively). While the intention-to-treat analysis also noted improved survival in the LT group, a subgroup analysis that included only patients with an R0 resection and N0 disease demonstrated no survival differences. The authors concluded that patients with resectable de novo pCCA should preferentially undergo resection [43].

Ethun et al. reported a multicentric prospective study based on data from the US Extrahepatic Biliary Malignancy Consortium. Interestingly, data from this study demonstrated that patients with unresectable pCCA treated with neoadjuvant therapy and LT had superior 5-year survival compared with patients treated with resection alone (64% vs. 18%). This finding remained significant after adjusting for tumor size, nodal status, and presence of PSC. LT patients more commonly received neoadjuvant treatment, while lymph node involvement was more frequently reported among patients who underwent resection. These data further confirmed the fundamental role of neoadjuvant therapy and the negative prognostic value of lymph-node involvement in the setting of pCCA [29].

Two studies reported much better survival among patients after resection. Nagino et al. reported on resected patients with Bismuth type IV pCCA and noted a 5-year OS of over 60% [46]. Comparable results were obtained by Ebata et al. among patients with Bismuth type IV pCCA who underwent resection and did not have lymph node involvement [47]. Due to these discordant data, a French randomized clinical trial called TRANSPHILL (NCT02232932) is ongoing. The study aims to compare outcomes among patients with resectable pCCA undergoing neoadjuvant chemoradiation followed by LT versus upfront surgery. The study is expected to be completed in 2024.

### 2.2. Intrahepatic Cholangiocarcinoma

Intrahepatic cholangiocarcinoma (iCCA) is the second most common primary liver malignancy. To date, the only curative option is surgery. Unfortunately, many patients are diagnosed at an advanced stage or have pre-existing liver cirrhosis that precludes resection [48]. Interest in LT as a treatment for iCCA started at the end of the last century. However, due to the reported poor OS (1- and 5-year: 50% and 25%, respectively), iCCA was considered a contraindication to LT [49,50]. Multifocal presentation, perineural invasion, infiltrative growth pattern, lack of neoadjuvant and adjuvant protocols, history of primary sclerosing cholangitis, and lymphovascular invasion were all factors related to disease recurrence and poor outcomes [51].

In recent years, there has been renewed interest in evaluating LT as a treatment for iCCA (Table 3). Patients with “very early” iCCA (i.e., single lesion <2 cm in diameter) considered unresectable due to underlying liver disease who underwent LT had similar post-LT survival compared with HCC patients meeting the Milan Criteria [52]. In this setting, Sapisochin et al. reported an acceptable 5-year OS of 65% and low recurrence rates [53,54]. A multicenter French study also reported acceptable outcomes among patients with cirrhosis who were transplanted for iCCA, even in single lesions of 2–5 cm as well as <2 cm (5-year OS: 65% and 69%; *p* = 0.40). Multivariable analysis demonstrated a correlation between tumor differentiation and recurrence [55]. In contrast, Lee et al. reported worse survival among patients with “very early” iCCA and cirrhosis versus HCC; 1-year OS was 63.6% for patients with iCCA versus 90.0% for patients with HCC, while 5-year OS was 63.6% for iCCA versus 70.3% for HCC. Patients with iCCA also had a higher incidence of recurrence (33.3% vs. 11.0%), poor tumor grade, and vascular invasion [56]. Gruttadauria et al. reported on an Italian series of 14 transplanted patients in a different study. Twelve iCCA cases were diagnosed after LT based on histologic findings, and two cases of unresectable iCCA were transplanted after neoadjuvant selective internal radiation therapy (SIRT) and a period of clinical observation. The two unresectable patients were alive after 19 and two months of follow-up [57].

Several series have been reported on the combination of neoadjuvant chemotherapy with LT for use in iCCA patients without cirrhosis. Lunsford et al. reported a case series of nine patients treated with gemcitabine-based neoadjuvant chemotherapy who subsequently underwent LT in stable disease or partial response. Six patients had favorable outcomes (OS at 1, 3, and 5 years: 100%, 83.3%, and 83.3%, respectively). Three patients had recurrence within five years after LT [60].

Recently, a multicentric US study compared the outcomes of patients with locally advanced pCCA and iCCA who received pre-LT gemcitabine plus cisplatin versus non-gemcitabine and cisplatin regimens. The gemcitabine group had improved OS (1- and 5-year: 100% vs. 75% and 75% vs. 63%, respectively) [62]. McMillan et al. reported on 18 patients transplanted after a neoadjuvant gemcitabine-based protocol plus locoregional or external beam radiation with at least six months of disease stability. Patients with locally advanced iCCA with a solitary tumor ≥2 cm in diameter or multiple tumors were included in the study cohort. The OS at 1, 3, and 5 years were 100%, 71%, and 57%, respectively. Seven patients who experienced recurrence were treated with aggressive protocols that included hepatic resection and adjuvant therapy regimens [63].

When comparing LT and resection, Kim et al. reported similar results in 66 patients (5-year OS = 36.1 vs. 34.7%; *p* = 0.53), but better results in LT patients compared with patients only receiving chemotherapy (5-year OS = 36.1 vs. 5.3%; *p* < 0.0001). LT may be an effective treatment among patients who are anatomically or physiologically unresectable [58]. Two studies compared the results of patients receiving resection versus combined (neo)adjuvant therapies plus LT. Hong et al. noted better OS among patients undergoing combined LT plus neoadjuvant and adjuvant systemic therapy versus patients treated with resection followed by adjuvant protocols [59]. In addition, a different series from the US National Cancer Database demonstrated comparable OS among patients receiving combined neoadjuvant therapy plus LT versus resection (median survival: 36.1 vs. 33.6 months; *p* = 0.57) [61].

In 2020, the University of Guangzhou developed a model to predict post-LT recurrence among patients with iCCA. Risk factors included tumor diameter, number of nodules, and CA 19-9 level. These three factors strongly predicted recurrence and death [65]. A multicentric study published by the Universities of Los Angeles and Milwaukee demonstrated excellent outcomes (5-year OS: 100%) among patients transplanted after chemotherapy or local therapy, irrespective of tumor size [64].

## 3. Liver Transplantation for Unresectable Colorectal Liver Metastases

Colorectal cancer represents one of the most common cancers worldwide, with half of patients developing liver metastases [66]. Among patients with resectable CRLM, resection is the only treatment option with potentially curative intent. However, only 10–15% of patients are eligible for this treatment approach [67], with 1–2% post-operative mortality and 50–60% 5-year survival rates [68]. In contrast, first-line treatment is palliative chemotherapy for patients with unresectable disease, and a subset of patients may also receive locoregional therapies. Among patients with unresectable CRLM, the median OS decreases to 5–10% [68,69,70]. Liver transplantation is an attractive treatment option for patients with unresectable liver metastases to improve their long-term survival.

The first reported experiences with LT for unresectable CLRM were disappointing [71]. In 1991, Mühlbacher et al. published the Vienna experience (N = 17) and reported a 5-year OS of 12% and a recurrence rate of 64% [72]. A North American series of 10 cases demonstrated a 5-year OS of 21% and a recurrence rate of 70% [73]. Given these findings, no further studies were published until the 2010s besides limited case reports. In 2010, an ad hoc analysis by the Oslo group used the ELTR data (N = 50) to revisit the early experience with LT for unresectable CRLM. In this analysis, 44% of graft loss/patient deaths were unrelated to tumor recurrence. The authors further compared their data with the UNOS transplant registry, noting a 1- and 5-year OS of 64% vs. 71% and 53 vs. 61%, respectively. Survival after LT for unresectable CRLM was concordant with UNOS liver transplant data for all transplant indications at that time [74]. These data led the authors to conclude that LT for CLRM in the current era would lead to improved outcomes following LT [67].

In 2013, the Oslo group published the results of the prospective Secondary Cancer (SECA)-I Study, a pilot study demonstrating that well-selected patients with unresectable CRLM confined to the liver could achieve excellent OS after LT [75]. SECA-I (N = 21) reported OS at 1, 3, and 5 years of 95%, 68%, and 60%, respectively. Unfortunately, the study also reported very high recurrence rates (65% after one year and 95% during the entire follow-up). A sub-analysis identified four different parameters affecting outcomes, creating the Oslo Criteria. The criteria include four factors: tumor diameter > 5.5 cm, carcinoembryonic antigen (CEA) > 80 μg/L, less than a two-year interval between primary resection and LT, and progressive disease at the time of LT. Using the Oslo Criteria, patients with from zero to one factor (n = 6) had a 100% survival at five years, while patients with all four factors (n = 5) all died within four years (*p* < 0.001) [75].

The follow-up SECA-II trial prospectively enrolled 15 patients with more strict selection criteria [76]. At a median follow-up of 36 months, patients had 1-, 3-, and 5-year OS of 100%, 83%, and 83%, respectively. Disease-free survival at 1, 2, and 3 years were 53%, 44%, and 35%, respectively. Recurrence was mainly in the form of slow-growing pulmonary metastases amenable to curative resection. SECA-II patients had significantly better prognostic factors than in the previous SECA-I study, demonstrating that improving selection criteria facilitated a much better 5-year OS for CRLM patients using LT. Confirming these findings, a study from the Compagnons Hépato-Biliaires Group retrospectively evaluated 12 patients transplanted between 1995 and 2015 in different European centers. These authors reported an OS of 83%, 62%, and 50% at 1, 3, and 5 years, respectively. Six patients had a recurrence with disease-free survival of 56%, 38%, and 38% at 1, 3, and 5 years, respectively (Table 4) [77]. A summary of several ongoing prospective studies aiming to validate these prior results is presented in Table 5.

The Oslo group compared 50 cases in the prospective SECA studies with 53 patients with resectable CRLM who had undergone pre-hepatectomy portal vein embolization (PVE), but otherwise had similar selection criteria. Interestingly, 28% of PVE patients did not proceed to resection, which significantly adversely impacted survival (5-year OS of 45% in resected vs. no surviving patients among those who did undergo resection). Among patients with low tumor load (<9 metastatic tumors or maximal tumor diameter ≤5.5 cm), patients who underwent LT had a 5-year OS of 72% vs. 53% among patients who underwent PVE plus resection (*p* = 0.08) [79], demonstrating a survival benefit for LT patients. Comparing LT and resection among patients with high tumor load (i.e., tumor burden score ≥9) demonstrated a 5-year OS benefit after LT (52.2% LT vs. 22.7% resection, p = 0.055) [80]. These data suggested that well-selected CRLM patients with high tumor load may benefit from LT with improved overall survival versus standard resection.

Another study from Oslo compared the SECA-I results with data from a randomized controlled trial (NORDIC-VII), in which first-line chemotherapy was used to treat liver-only metastatic patients [81]. When comparing the 21 SECA-I patients with 47 NORDIC-VII cases, a dramatic difference in OS was observed (5-year rate: 56% vs. 9%, respectively). In contrast, the median disease-free time was similar in the two groups (8–10 months), yet there were different metastatic patterns of relapse/progression observed in the two groups. Specifically, relapse in the SECA-I group was more often detected as small, slowly growing lung metastases [81].

The Oslo group evaluated the use of adjuvant chemotherapy after LT. The authors investigated 43 transplanted patients and reported 33 cases of relapse. Twenty-three patients received some form of post-recurrence chemotherapy. Chemotherapy after LT was either combination regimens including 5-fluorouracil/folinic acid with irinotecan or oxaliplatin or monotherapy consisting of either capecitabine or irinotecan. A total of 9 of 23 patients received chemotherapy combined with an anti-EGFR antibody, while 6 received bevacizumab. No evidence of graft rejection was reported, but 19 out of 23 patients (83%) reported one or more grade 3–4 toxicity events. The most adverse effects included impaired bone marrow function, diarrhea, or mucositis. Median OS from the start of palliative chemotherapy after LT was 13 months (range = 1–60). No cases of grade 5 toxicity were reported, confirming LT patients can tolerate chemotherapy after transplant with extended survival compared with the best supportive care [82]. In many relapse cases, tumor growth is indolent and manageable with relative safety of using post-LT chemotherapy. When comparing LT with upfront chemotherapy or resection, liver transplant has demonstrated a benefit in terms of overall survival.

## 4. Expanding the Donor Pool

A significant risk factor for dropout and recurrence for any oncological indication for liver LT is a prolonged waiting time between the end of neoadjuvant therapy and transplantation. Multiple approaches to expanding the donor pool have been studied. One strategy to bypass a long waiting time after neoadjuvant therapy is living donor liver transplantation (LDLT). Tan et al. compared 73 cases of LDLT for pCCA with 173 LDLT performed for other indications. LDLT for pCCA was associated with nonstandard arterial or portal vein reconstruction, roux-en-y choledochojejunostomy, and more cases of late hepatic artery and portal vein complications. Anastomotic biliary complications arose with no differences between the two groups. The five-year OS among patients with pCCA was 66.5% (75.9% in PSC and 47.5% in de novo pCCA), with an incidence of tumor recurrence of 12.3%. Therefore, the authors concluded that late vascular complications were more common after LDLT for pCCA than otherwise, but these complications did not adversely affect long-term survival [31].

A prospective study from North America reported ten cases of LDLT for unresectable CRLM, in which eight right lobe grafts and two left lobe grafts were directly implanted. Live donation was safe, with only one donor experiencing Clavien–Dindo III complication after donation. As for the recipients, three recurrences were reported after LDLT with recurrence-free and OS of 62% and 100% at 1.5 years after live-donor LT, respectively [83]. The Oslo group reported a new technique called RAPID (Resection And Partial Liver Segment 2/3 Transplantation With Delayed Total Hepatectomy) for patients with unresectable CRLM. In this technique, a left hepatectomy was performed, an auxiliary small left lateral split graft was implanted, and the right portal vein was ligated to induce hypertrophy. After three weeks, the hepatectomy was completed when the implanted graft reached a sufficient volume [84]. The Tubingen group expanded on the possibilities related to this procedure by reporting the first successful living donor RAPID procedure [85].

Given the ongoing shortage of available organs, it is critical to perform studies evaluating the use of marginal deceased donors for LT for oncological indications. This topic was investigated in the SECA-II arm D study and a study from the Karolinska Institute. The SECA-II arm D study was a prospective single-arm study that evaluated patients with synchronous unresectable CRLM who had undergone resection of the primary tumor and received chemotherapy. These patients were ineligible for SECA-II arms A–C given their advanced disease. These 10 patients with extensive liver disease (number and size of lesions) underwent LT from extended criteria donor grafts; median disease-free and overall survival were 4 and 18 months, respectively [78]. The study from the Karolinska Institute used retrospective data to evaluate (1) the potential for declined donors who could have been acceptable as extended criteria donors, and (2) patients with unresectable CRLM as potential liver transplant recipients. The aim was to evaluate the potential increase in the donor pool with the inclusion of extended criteria donors and whether the use of a marginal donor was associated with an acceptable risk–benefit ratio among patients with unresectable CRLM. In the analysis, the use of extended criteria donors would have increased the acceptable potential donor pool by 6–18%. The authors then evaluated patients with unresectable CRLM and reported that a small subset of patients would have been eligible for LT. Median OS for patients with unresectable CRLM, yet eligible for LT, was higher than those with unresectable CRLM and ineligible for LT (18 vs. 12 months, *p* = 0.037). As such, using extended criteria donors in well-selected patients with unresectable CRLM may be reasonably considered [86].

Organ shortage also implies the need to consider the relevant inherent ethical aspects correlated with the decision to transplant a patient with an uncommon oncological indication. In fact, when a graft is dedicated to a patient with this indication, it is not contemporaneously used for transplanting a cirrhotic patient. Therefore, potential harm in the list of non-tumoral patients must be considered. In this light, several studies have been published exploring the impact of transplanting HCC patients, with particular attention paid to the concept of “transplant benefit” [87,88]. Unfortunately, no specific studies have explored this relevant aspect in the setting of the new indications for transplantation. Further studies are needed with the intent to clarify this aspect.

## 5. Conclusions

There has been growing interest in transplant oncology over the last decade. Advances in neoadjuvant and adjuvant therapies coupled with ongoing studies on the use of LT in CCA and CRLM may facilitate outcomes following liver transplants performed for oncologic reasons.

The advances in immunosuppression represent further discussion in this specific setting. In the case of transplantation for HCC, it is well known that immunosuppression plays a relevant role in the risk of recurrence and that the minimization or the management of rejection should be correlated with a reduced or increased risk of tumor recurrence [89,90]. In recent years, the everyday use of drugs correlated with a reduced carcinogenic risk, such as everolimus, has modified the post-LT overall tumor risk [91]. The suboptimal results observed in the first series of LT for CCA and CRLM were linked with higher doses of immunosuppressants used in that period. Nowadays, a “tailored” immunosuppressive approach, involving the anti-angiogenetic drug everolimus, is thought to be the cornerstone of therapy in this transplant oncology setting and predicted to become the standard approach.

In light of the significant heterogeneity of the pre-LT conditions (i.e., lymph node status, resectability, tumor size, tumor biology) and absence of standard neoadjuvant regimens, mainly in the setting of CCA, it appears evident that the most crucial aspect in aLT for these uncommon tumor pathologies is accurate patient selection.

For locally advanced unresectable pCCA, neoadjuvant chemoradiation followed by LT provides reasonable long-term OS in a well-selected population. Although iCCA is still considered a contraindication to LT in most centers worldwide, recent studies have demonstrated acceptable outcomes for “very early” tumors in cirrhotic patients and patients receiving neoadjuvant therapies prior to LT. The initial results of LT for unresectable CRLM have demonstrated improvements in OS, although the risk of recurrence remains high, and some of these patients still might benefit from other approaches such as tumor ablation and radio- and chemo-embolization.

For unresectable CCA and CRLM management, several prospective protocols are forthcoming to elucidate the impact of LT relative to alternative therapies such as resection, systemic, and/or locoregional therapy in terms of long-term survival. Advances in diagnosis, treatment protocols, and donor-to-recipient matching are needed to better define the oncological indications for transplantation. Prospective, multicenter trials studying these advances and their impact on outcomes are now required.

## Figures and Tables

**Table 1 medicina-59-00568-t001:** Characteristics and findings of studies focused on liver transplantation and unresectable perihilar cholangiocarcinoma.

Author	Year	Ref.	Country	N	Transplanted	Neoadjuvant Therapy	OS %			DFS %		
							1-Year	3-Year	5-Year	1-Year	3-Year	5-Year
Heimbach	2004	[22]	US	106	65 (61%)	Yes	91	-	76	-	-	-
Rea	2005	[11]	US	71	38 (54%)	Yes	92	82	82	0 *	5 *	12 *
Robles	2007	[23]	Spain	66	10 (15%)	No	80	60	37	-	-	-
Kaiser	2008	[24]	Germany	47	47 (100%)	No	61	31	22	-	-	-
Seehofer	2009	[15]	Germany	16	16 (100%)	No	63	-	38	-	-	-
Rosen	2012	[25]	US	136	136 (100%)	Yes	92	81	74	-	-	-
Darwish	2012	[26]	US	287	216 (75%)	Yes	-	68 ** (§)	53 (§)	-	78 **	65
Mantel	2016	[27]	Europe	173	105 (61%)	Yes	-	-	32	-	-	-
Dondorf	2018	[28]	Germany	22	22 (100%)	No	89	36	29	78	32	24
Ethun	2018	[29]	US	304	70 (23%)	Yes	-	72	64	-	-	-
Zaborowski	2020	[30]	Ireland	37	26 (70%)	Yes	81	69	55	76	63	52
Tan	2020	[31]	US	247	74 (30%)	Yes	85	67	56	-	-	-

* recurrence rate; ** 2-year; (§) intention-to-treat survival. Abbreviations: Ref., reference; N, number; OS, overall survival; DFS, disease-free survival; NA, not available.

**Table 2 medicina-59-00568-t002:** Studies exploring neoadjuvant radio-chemotherapy followed by transplantation vs. upfront resection for perihilar cholangiocarcinoma.

Study	Year	Ref.	Country	LT (n)	Resection (n)	Neoadjuvant Therapy	Staging Surgery	Median Follow-Up (Months)	LT vs. Resection	
									R0	3-Year OS %
Robles	2007	[23]	Spain	10	23	No	Yes	NA	NA	59 vs. 60
Hidalgo	2008	[44]	UK	12	44	No	Yes	22	9/12 vs. 20/44	41 vs. 43
Kaiser	2010	[45]	Germany	7	7	No	No	32	6/7 vs. 6/7	71 vs. 71
Croome	2015	[43]	USA	54	99	Yes (54/54)	Yes	43	54/54 vs. 90/99	71 vs. 53
Ethun	2018	[29]	USA	41	191	Yes (39/41)	Yes	23	36/41 vs. 134/191	72 vs. 33

Abbreviations: Ref., reference; LT, liver transplantation; n, number; R, resection; OS, overall survivals; NA, not available.

**Table 3 medicina-59-00568-t003:** Characteristics and findings of the studies focused on liver transplantation and intrahepatic cholangiocarcinoma.

Author	Year	Ref.	Country	Study Group (n)	5-Year OS %	5-Year Recurrence %
**Cirrhotic liver**						
Sapisochin	2014	[53]	Spain	n = 8 single <2 cm n = 21 single ≥2 cm or multiple	73 34	0 42
Sapisochin	2016	[54]	Spain	n = 15 < 2 cm n = 33 single ≥2 cm or multiple	65 33	18 61
Lee	2018	[56]	US	n = 44 incidental	64	33
De Martin	2020	[55]	France	n = 10 lesion(s) <2 cm n = 14 lesion(s) 2–5 cm	65 69	24 28
Kim	2022	[58]	US	n = 66 localized non-resectable	36	NA
**Non-cirrhotic liver**						
Hong	2011	[59]	US	n = 25 locally advanced (n = 9 neoadjuvant therapies)	34	62
Lunsford	2018	[60]	US	n = 9 advanced (all CHT)	83	50
Gruttadauria	2021	[57]	Italy	n = 7 incidental n = 2 unresectable	69 NA	NA NA
Hue	2021	[61]	US	n = 74 unresectable	41	NA
Abdelrahim	2022	[62]	US	n = 10 advanced (all CHT) n = 8 advanced (all CHT)	75 63	NA NA
McMillan	2022	[63]	US	n = 18 advanced (all CHT + locoregional or external beam radiation)	57	38
Ito	2022	[64]	US	n = 31 locally advanced	100	NA

Abbreviations: Ref., reference; OS, overall survival; US, United States; NA, not available; CHT, chemotherapy.

**Table 4 medicina-59-00568-t004:** Characteristics and findings of the studies focused on liver transplantation and colorectal liver metastases.

Author	Year	Ref.	Period	N	Study	5-Year OS	Recurrence
Mühlbacher	1991	[72]	1982–1991	25	-	12%	64%
Penn	1991	[73]	NA	10	-	21%	70%
Foss	2010	[67]	1983–1994	50	-	18%	NA
Hagness	2013	[75]	2006–2011	21	SECA-I	60%	35% (1-yr)
Toso	2017	[77]	1995–2015	12	-	50%	56% (1-yr)
Dueland	2020	[76]	2012–2016	15	SECA-II	83%	14 mo (mean)
Smedman	2020	[78]	2014–2018	10	SECA-II arm D	18 mo (mean)	4 mo (mean)

Abbreviations: Ref., reference; N, number; OS, overall survival; NA, not available; SECA, Secondary Cancer; mo, months.

**Table 5 medicina-59-00568-t005:** Current studies evaluating liver transplantation in colorectal liver metastases.

NCT Number	Study Name	Year	Type	Patients	Country	Study Aims
02215889	No	2014–2028	Intervention	20	Norway	Single arm (segment 2, 3 partial LT)
02597348	TRASMET	2015–2027	RCT	90	France	LT plus chemotherapy vs. chemotherapy
02864485	No	2016–2023	Non-RCT	20	Canada	Single arm (chemotherapy followed by LDLT)
03231722	COLT	2017–2024	Multicenter non-RCT	432	Italy	LT vs. chemotherapy (parallel arm in TRIPLETE trial)
03488953	LIVER-TWO-HEAL	2018–2023	Intervention	40	Germany	Single arm (LDLT with two-staged hepatectomy)
03494946	SECA III	2016–2027	RCT	25	Norway	LT vs. chemotherapy
04161092	SOULMATE	2020–2030	Multicenter RCT	45	Sweden	LT (extended criteria graft) vs. best alternative therapy
04616495	TRASMETIR	2021–2028	Multicenter non-RCT	30	Spain	Single arm (LT)
04865471	RAPID-Padua	2020–2025	Non-RCT	18	Italy	Single arm (segment 2, 3 partial LT)
04742621	No	2020–2034	Non-randomized, single-arm, pilot registry	20	US	LT
04870879	MELODIC	2020–2025	Multicenter non-RCT	18	Italy	LT vs. chemotherapy (matched cohort)
04874259	No	2022–2026	Non-RCT	20	Korea	Single arm (LDLT)
04898504	EXCALIBUR 1 + 2	2021–2026	Three-arm parallel RCT	45	Norway	2nd line chemotherapy + HAI-floxuridine or LT versus 2nd line chemotherapy alone
05175092	No	2022–2030	Non-RCT	50	US	Single arm (chemotherapy followed by LDLT)
05185245	No	2021–2030	Non-RCT	20	Italy	Single arm (deceased and LDLT)
05186116	LIVERMORE	2022–2032	Non-RCT	25	Italy	LDLT
05248581	No	2019–2027	Non-randomized, single-arm, pilot registry	25	US	LDLT
05398380	No	2022–2026	Non-RCT	35	Spain	Single arm (LT)

Abbreviations: NCT, number on ClinicalTrials; LT, liver transplantation; RCT, randomized controlled trial; LDLT, living donor liver transplant; HAI, hepatic artery infusion.

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
