# Peer review of "New Insights in the Setting of Transplant Oncology"

_medicina, 2023, doi:10.3390/medicina59030568_

Round 1

Reviewer 1 Report

This is an interesting overview regarding LT in the oncological context. I only have a few remarks:

1. „Interestingly, Lehrke et al. reported that 52% of patients undergoing LT for
clinically diagnosed pCCA had no evidence of disease on final pathology [33]“

This is somewhat misleading, as the cited study reported on results of pretreated CCA, but the study is mentioned in a paragraph about difficulty of initial right diagnosis.

2. To give context to survival rates, it might be helpful to mention those from patients transplanted for HCC.

3. It might be interesting, to mention disease-free survival (if available).

4. The changing role and management of post-LT immunosuppression might explain worsened outcome from early years. A brief overview could help.

5. Organ shortage and the inherent ethical aspects due to shortage could be mentioned.

4. In my opinion, the Discussion/conclusion is a little too short. Given the heterogeneity of pre-LT conditions (lymph node status in pCCA, resectability, tumor size in iCCA, tumor biology for CRLM) and no standard neoadjuvant regimens for CCA, it is difficult to say, which patient is eligible for LT in this context and patient selection might be the most crucial aspect (as highly-effective oncological regimens for CCA still do not exist). Also, patients with unresectable CRLM still might profit other approaches (ablation, TACE). From my perspective, LT for CRLM and CCA remains without solid foundation and should be regularly conducted only in the setting of clinical trials.

Author Response

This is an interesting overview regarding LT in the oncological context.

Authors’ response: We thank the Reviewer for the opportunity to improve the quality of the study and for giving us the possibility of being reconsidered for publication.

  1. “Interestingly, Lehrke et al. reported that 52% of patients undergoing LT for
    clinically diagnosed pCCA had no evidence of disease on final pathology [33]”.

This is somewhat misleading, as the cited study reported on results of pretreated CCA, but the study is mentioned in a paragraph about difficulty of initial right diagnosis.

Authors’ response: We agree with the Reviewer. A misleading interpretation of the data reported in the paper has been made. The study by Lehrke et al. reported the complete response after pre-LT treatments and not an erroneous diagnosis before LT. We entirely removed the sentence according to the suggestions of the Reviewer.

  1. To give context to survival rates, it might be helpful to mention those from patients transplanted for HCC.

Authors’ response: We agree with the Reviewer. At the beginning of the article, we added the most recent data on OS and recurrence rates in the setting of HCC.

  1. It might be interesting, to mention disease-free survival (if available).

Authors’ response: We agree with the Reviewer. Unfortunately, in many cases, the DFS rates were not reported in detail. In many cases, only the number of recurrences was generically reported without any temporal clarification. Nevertheless, we checked all the studies reported in the tables. Where available, we reported the available data. We added in Table 1 a new column with the DFS data available at 1, 3, and 5 years after LT.

  1. The changing role and management of post-LT immunosuppression might explain worsened outcome from early years. A brief overview could help.

Authors’ response: According to the suggestions of the Reviewer, we added in the conclusive part of the paper the following sentence:

The advances in immunosuppression represent further discussion in this specific setting. In the case of transplantation for HCC, it is well known that immunosuppression plays a relevant role in the risk of recurrence and that the minimization or the management of rejection should be correlated with a reduced or increased risk of tumor recurrence [81,82]. In recent years, the everyday use of drugs correlated with a reduced carcinogenic risk like the everolimus has modified the post-LT overall tumor risk [83]. The worse results observed in the first series of LT for CCA and CRLM should also be relevantly conditioned from higher doses of immunosuppressants used in that period. Nowadays, a “tailored” immunosuppressive approach, comprehending as a cornerstone of the therapy the anti-angiogenetic drug everolimus, represents the standard approach in this transplant oncology setting.

  1. Organ shortage and the inherent ethical aspects due to shortage could be mentioned.

Authors’ response: We agree with the Reviewer. In light of these considerations, in the last part of the subchapter entitled “Expanding the Donor Pool”, we added the following sentence:

Organ shortage also implies the necessity to consider the inherent ethical aspects correlated with the decision to transplant a patient with an uncommon oncological indication. In fact, when a graft is dedicated to a patient with this indication, it is not contemporaneously used for transplanting a cirrhotic patient. Therefore, potential harm in the list of non-tumoral patients must be considered. In this light, several studies have been published exploring the impact of transplanting HCC patients, with particular attention to the concept of “transplant benefit” [81,82]. Unfortunately, no specific studies have explored this relevant aspect in the setting of the new indications for transplantation. Further studies are needed with the intent to clarify also this aspect.

  1. In my opinion, the Discussion/conclusion is a little too short. Given the heterogeneity of pre-LT conditions (lymph node status in pCCA, resectability, tumor size in iCCA, tumor biology for CRLM) and no standard neoadjuvant regimens for CCA, it is difficult to say, which patient is eligible for LT in this context and patient selection might be the most crucial aspect (as highly-effective oncological regimens for CCA still do not exist). Also, patients with unresectable CRLM still might profit other approaches (ablation, TACE). From my perspective, LT for CRLM and CCA remains without solid foundation and should be regularly conducted only in the setting of clinical trials.

Authors’ response: According to the suggestions of the Reviewer, we slightly increased this part, adding the considerations on immunosuppression and integrating the comments of the Reviewer in part.

Reviewer 2 Report

The authors address two areas of transplant oncology as related to liver transplantation. In these two areas the authors present comprehensive data.   I found the review to be comprehensive with no gaps in data reviewed.  

Author Response

The authors address two areas of transplant oncology as related to liver transplantation. In these two areas the authors present comprehensive data.   I found the review to be comprehensive with no gaps in data reviewed. 

Authors’ response: We thank the Reviewer for the positive comments and the opportunity to consider our article feasible for publication.
